# Monitoring Changes in the Volatile Compounds of Tea Made from Summer Tea Leaves by GC-IMS and HS-SPME-GC-MS

**DOI:** 10.3390/foods12010146

**Published:** 2022-12-27

**Authors:** Jiyuan Xu, Ying Zhang, Fei Yan, Yu Tang, Bo Yu, Bin Chen, Lirong Lu, Liren Yuan, Zhihua Wu, Hongbing Chen

**Affiliations:** 1State Key Laboratory of Food Science and Technology, Nanchang University, Nanchang 330047, China; 2College of Food Science and Technology, Nanchang University, Nanchang 330031, China; 3Kylin (Shandong) Pharmaceutical Technology Co., Ltd., Jinan 250109, China; 4Sino-German Joint Research Institute, Nanchang University, Nanchang 330047, China; 5Riantea Ltd., Nanchang 330100, China

**Keywords:** aroma compounds, Fuliangzhong, summer tea, head space-solid phase micro-extraction gas chromatography-mass spectrometry (HS-SPME-GC-MS), gas chromatography-ion mobility spectrometry (GC-IMS), relative odor activity value (ROAV)

## Abstract

Compared with spring tea, summer tea has the advantages of economy and quantity. However, research on the aroma characteristics of summer tea is currently limited. In this study, summer fresh tea leaves (*castanopsis. sinensis*, cv. Fuliangzhong) (FTLs) were processed intoblack tea (BT) and green tea (GT). The changes in the volatile compounds during the tea processing were quantified using gas chromatography-ion mobility spectrometry (GC-IMS) and head space-solid phase micro-extraction gas chromatography-mass spectrometry (HS-SPME-GC-MS), and then analyzed on the basis of relative odor activity value (ROAV). Results showed low amounts of flavor compounds, such as linalool oxides, geraniol, and sulcatone, were found in FTLs, but after processing, high amounts of the same in BT and GT. Summer BT and GT contained characteristic compounds similar to spring tea, including linalool, geraniol, (E,E)-2,4-decdienal, β-ionone, methyl salicylate, geranyl acetone, and decanal. All these compounds have high content and ROAV values, which give the same flavor to summer teas as spring tea. This study confirmed that summer fresh tea leaves were suitable to produce black and green tea with good flavor. Monitoring changes in aroma compounds by GC-IMS coupled with GC-MS, the quality of summer tea is expected to be promoted towards the quality of spring tea by improving processing methods for valuable-tea production.

## 1. Introduction

As one of the top three drinks in the world [1], tea has a huge consumer market, and the annual tea output is about six million tons, about half of which is produced in China. Spring tea constitutes less than half (40–45%) of China’s output. Therefore, summer and autumn tea leaves are of particular importance.

Tea made from fresh tea leaves plucked in spring has long been believed to be better quality than tea made from tea leaves plucked in summer or autumn. Thus, tea made with spring tea leaves, with a mellow and umami taste, and a floral and fragrant odor, is considerably more expensive than tea made from fresh tea leaves plucked in summer and autumn [2]. Compared with spring tea, summer tea has significantly higher amounts of catechins and other non-volatile compounds, which contribute to bitter and astringent tastes in tea infusions [3]. Therefore, most fresh tea leaves collected during the summer are discarded directly. However, summer tea production has a large output and is less expensive. Summer tea leaves can be made into green, black, oolong, dark, white, and yellow tea because of their special flavors and unique or characteristic odors obtained through different manufacturing processes [4]. Two types of tea are widely available in the global market: nonfermented (green tea) and completely fermented (black tea) [5,6].

Tea aroma is the soul of tea and is one of the key factors that attracts consumers. Volatile compounds occur in minute quantities in tea, accounting for only 0.01–0.05% of the total dry mass of tea. More than 600 kinds of these compounds, such as linalool and its oxides, geraniol, and β-ionone have been detected in tea products, contributing greatly to tea quality [7]. At present, the research on tea aroma is mainly focused on spring tea, and relatively little research has been done on summer tea [8]. Previous research shows that the summer tea was suitable to produce black or green tea with good flavor [1,9]. However, these studies only used summer tea to make one type of tea and used one method to detect its aroma substances, so the utilization and detection results for summer tea are not comprehensive.

GC-IMS is a detection technology that combines ion mobility spectroscopy and gas chromatography [10] and can analyze the test results quickly, simply, intuitively, and accurately. It has been widely used in many fields, such as food flavor analysis [11] and quality inspection [12]. GC-MS has also been widely used in the separation and identification of volatiles in food [13]. Head space-solid phase micro-extraction (HS-SPME) is a volatile extraction technology that is superior to traditional extraction methods and is efficient, fast, and solvent-free [14]. This method is commonly used to extract the volatile compounds of various types of foods, such as wine [15], meat products [16], and tea [17]; it shows good repeatability, sensitivity and selectivity. In recent years, GC-MS and GC-IMS are often used in tea flavor analysis [17,18]. Therefore, GC-MS and GC-IMS can be used in analyzing changes in volatile compounds in tea made from summer fresh tea leaves (*castanopsis. sinensis*, cv. Fuliangzhong).

Flavor compounds can be monitored through GC-MS and GC-IMS. The main purpose of the current study is to investigate the aroma profiles of black tea (BT) and green tea (GT) produced with summer tea leaves. By bridging flavor change with substances, we can promote the quality of summer tea towards spring tea. In the present study, the summer fresh tea leaves (*castanopsis. sinensis*, cv. Fuliangzhong) were made into black tea (BT) and green tea (GT). After different treatments, changes in volatile compound were quantified by GC-MS and GC-IMS and analyzed by relative odor activity value (ROAV).

## 2. Materials and Methods

### 2.1. Reagents and Materials

Summer fresh teas (*castanopsis. sinensis*, cv. Fuliangzhong) were manufactured by a tea master with more than 30 years of experience in tea production in the region of Yellow Mountain City (Fuliang, China) according to the traditional processing of fresh tea leaves into final teas. The processing method is as follows:

Green tea preparation: The fresh tea leaves measured about 3 cm, and were spread out at 20 °C for 4 h. Wilting leaves were fixed in an electric frying pan at about 220 °C. Killed leaves were rolled on a rolling machine according to the method of “light rolling for 10 min→heavy rolling for 5 min→light rolling for 5 min”. Rolled leaves were unblocked and placed in an oven and dried at 85 °C for 60 min. 

Black tea preparation: The fresh tea leaves were spread until the moisture content reached 60~62%. Wilting leaves were rolled on rolling machine according to the method of “light rolling for 30 min→medium rolling for 15 min→heavy rolling for 20 min→light rolling for 5 min”, and then the rolled leaves were unblocked and placed in a fermentation room with a temperature of 30 °C and a relative humidity of 95% for 4 h. The tea was dried in an oven with an initial drying temperature of 120 °C for 15 min, and the oven was properly cooled, and then the tea was fully dried at 90 °C for 60 min. 

The fresh tea leaves and the prepared black and green teas were collected and stored at −20 °C until use. Accordingly, the fresh tea leaves and final product black and green teas from full-fire processing were named FTL, BT and GT, respectively.

### 2.2. HS-GC-IMS Analysis Methods

Analyses of the teas were performed using the GC-IMS (gas chromatography-ion mobility spectrometry) as described by Yang et al. [19], with slight modifications. Specifically, tea samples (1.0 g) were weighed and placed in a headspace vial with a volume of 20 mL. The headspace glass vials were incubated on a heated plate (80 °C) for 15 min. After incubation, 200 μL of headspace was automatically injected into the injector under splitless injection mode with a syringe at 85 °C. GC was performed with a MXT-5 capillary column (15 mm × 0.53 mm × 1 μm; column temperature: 60 °C) to separate volatile compounds and coupled to IMS at 45 °C. Nitrogen (99.999% purity) was used as the carrier gas; the drift gas (nitrogen gas) was set at 150 mL/min. C4–C9 n-ketones (Sinopharm Chemical Reagent Beijing Co., Ltd., Beijing, China) were used as external references to allow the retention index (RI) of the detected volatiles to be calculated under the same chromatographic conditions.

GC-IMS Data analysis was carried out using functional software, Laboratory Analytical Viewer analysis software, and three plug-ins: Reporter, Gallery Plot, and Dynamic PCA, and GC × IMS Library Search (G.A.S. Gesellschaft fur analytische Sensorsysteme embH, Dortmund, Germany). Built-in NIST 2014 gas-phase retention index database and G.A.S IMS migration time database were used for two-dimensional qualitative analysis.

### 2.3. HS-SPME-GC-MS Analysis Methods

Sample pretreatment: different tea samples (1.0 g) were placed into a 20 mL SPME vial, then combined with 10 mL at 80 °C of ultrapure water and equilibrated at 80 °C for 20 min. Then the SPME DVB/CAR/PDMS-coated SPME fiber (2 cm) was inserted into the balanced sample vial, with a constant-temperature water bath at 80 °C for 40 min; the SPME fiber was thermally desorbed at 250 °C for 5 min in a splitless injection port of the GC for analysis. 

GC-MS conditions were according to the procedure described by Zhang et al. [20], with modifications; an Agilent 7890B-7000D model GC-MS system (Agilent, USA) with a fused-silica capillary column (DB-35MS, 30 mm × 0.25 mm × 0.25 μm; Agilent, USA) was used for the analysis of tea volatiles. The initial temperature was 50 °C, retained for 2 min, rising to 90 °C at the rate of 4 °C/min and held for 3 min, then heated to 150 °C at the rate of 3 °C/min and held for 5 min, finally heated to 250 °C at the rate of 5 °C/min and held for 5 min. Helium was used as the carrier gas, at a flow rate of 1.0 mL/min with the splitless GC inlet mode. The MS fragmentation was performed by electronic impact (EI) mode (ionization energy, 70 eV; source temperature, 230 °C). The quadrupole temperature was 150 °C. The acquisition was in full-scan mode with a mass acquisition range of 35–550 *m*/*z*. 

The obtained chromatograms were analyzed by determining the peak areas, retention times, spectra and base peaks on the chromatograms, and then each peak was examined by referring to the characteristic mass spectra of the compounds listed on the National Institute of Standards and Technology (NIST). The relative contents of each compound in the volatile compounds of tea samples were obtained by the area normalization method. If the similarity index is greater than 80, it is considered that the compound exists in the sample.

### 2.4. Identification of Key Aroma Compounds

On the basis of relative quantification, the threshold value of each aroma compound in water was checked, with reference to the research method of Ma et al. [21]. Then, the relative odor activity value (ROAV) of each aroma compound was calculated as follows:$${\mathrm{ROAV}}_{i}\;\approx \;\frac{C_{i}}{C_{\max}}\times \frac{T_{\max}}{T_{i}}\times 100$$

In the formula, C_i_ is the relative content of the aroma compound in tea (%); T_i_ is the aroma threshold of compound in water (μg/kg); C_max_ and T_max_ represent the relative content and aroma threshold of the compound that contributes the most to the overall flavor of the sample. For all compounds, ROAV ≤ 100; the larger the ROAV value, the greater the contribution of the component to the overall flavor of the sample. The compounds where ROAV ≥ 1 are identified as the key flavor compound of the analyzed sample, the 0.1 ≤ ROAV < 1 compounds are believed to be important modifiers on the overall flavor of the sample.

### 2.5. Statistical Analysis

The data was entered and sorted by Microsoft Excel 2010, and Origin 2018 software was used for mapping analysis. As for GC-IMS, the instrumental analysis software includes Laboratory Analytical Viewer (LAV version 2.0.0, G.A.S., Dortmund, Germany), Gallery Plot, and Reporter, as well as GC-IMS Library Search, which can be used for sample analysis from different angles. 

## 3. Results and Discussion

### 3.1. Volatile Compounds Identified by GC-IMS

#### 3.1.1. Analysis of the Topographic Plots of Volatile Components in Teas by GC-IMS

The GC-IMS technique, with high separation efficiency, fast response of ion mobility spectra, and high sensitivity, was used to analyze the volatile compounds of tea leaves after different treatments [10,22]. The two-dimensional top view of GC-IMS (Figure 1A) can visually compare the composition of volatile substances among different samples. White indicates lower intensity, and red indicates higher intensity. Compared with the FTL, the number of flavor substances in BT and GT had increased, and the concentration of most flavor substances had also increased.

A difference-comparison model was used in comparing the volatile varieties of the tea samples (Figure 1B). If the volatile compounds were consistent, the background after deduction was white, while red indicated that the concentration of the substance was higher than the reference, and blue indicated that the concentration of the substance was lower than the reference. The topographic plot of the fresh tea leaves (FTLs) was selected as a reference, and the topographic plot of the other samples was subtracted from the reference. Each point represents one compound. Compared with the number of FTLs, the number of volatile compounds in BT and GT increased, and the concentrations of most volatile compounds also increased [23]. The changes in BT are more obvious than those in GT.

#### 3.1.2. Differences of Volatile Compounds in Teas by GC-IMS

The content of each volatile compound is shown in Table 1. A total of 75 volatile compounds were detected in the three samples. The same series of compounds were used in analyzing the changing regularities of volatile compounds in tea samples subjected to different treatments. Aldehydes, alcohols, and ketones were the major volatiles, the proportions were 36.12%, 18.04%, and 25.23%, respectively (FTL); 38.27%, 26.83%, and 19.68%, respectively (BT); and 45.26%, 22.03%, and 18.10%, respectively (GT), as shown in Figure 2. In contrast to FTLs, BT and GT have higher levels of aldehydes and alcohols, and lower levels of ketones. BT had the highest content of alcohols, while GT had the highest content of aldehydes.

To show the changing regularities and relative content of the volatile compounds of the FTLs in different treatments, we performed gallery plot analysis as a fingerprinting technique [24]. The differences in volatile compounds in each sample are presented in Figure 3. Significant differences in volatile compounds observed in GC-IMS were marked as fingerprints. The content of the compounds varied with signal intensity. Changes in the volatile compounds in the processed teas were analyzed by the gallery plot analysis, which revealed that each sample had its own characteristic aroma components (the different colored boxes in Figure 3). 

After the FTLs were processed into BT, the total content of the volatile compounds increased; 57 individual compounds increased, and only 18 decreased. The compounds affected the flavor and quality of BT, producing a sweet taste, and floral and fruit aromas. Although the content of dimethyl sulfide was significantly reduced in BT (from 4.67% to 0.67%), it generated a delicate scent at low concentration [17], thus exerting a positive effect on the quality of BT. 

The main aroma compounds in BT, including (E)-2-hexenol, linalool and its oxides, benzaldehyde, hexanal, benzenecetaldehyde, methyl salicylate, and furfural, showed increased content (shown in Table 1), which is consistent with previous research results [25]. One of the important compounds that determines the tender aroma of BT is (E)-2-hexenol, with green, fruit, and vegetable odors [23], which increased in content more than tenfold, and the relative content was high. Owing to withering, the content of (E)-2-hexenol can increase more than tenfold than that in fresh leaves [6]. 

As an important source of flower and fruit aromas in BT, linalool showed an obviously increased level. In BT, the linalool content increased from 0.69% to 1.67%. Linalool is the characteristic aroma compound of Keemun black tea (containing 2.54%), and is mainly formed from the hydrolysis of glucoside during fermentation [26]. Therefore, the degree of glucoside hydrolysis can be increased by increasing the fermentation time or temperature. This approach increases linalool content.

Phenylacetaldehyde has a floral odor, and benzaldehyde may contribute to the burnt sugar odor in tea. In spring tea, phenylacetaldehyde and benzaldehyde are mainly produced by the degradation and oxidation of amino acids and reducing sugars [27]. These two flavor compounds were detected in summer tea, producing a floral and sweet-caramel aroma, and were also detected at high concentrations in Keemun black tea [26]. 

After the FTLs were processed into GT, the content of total volatile compounds increased. Compared with the content of FTL, the content of 50 volatile compounds increased, whereas the content of 25 volatile compounds decreased in GT. These compounds affect the flavor and quality characteristics of green tea through their combined effects, contributing to the unique odor of GT. Dimethyl sulfide was significantly reduced (from 4.67% to 2.38%), and showed a delicate scent at low concentrations [17]. It is the main aroma compound in Laoshan green tea [28]. 

The content of main aroma compounds of GT, including furfural, nonanal, hexanal, (E,E)-2,4-heptadienal, linalool and its oxides, pentanal, sulcatone, and ethyl acetate, increased (Table 1). In GT production, furfural, with sweet, bready, and caramel odors, showed the largest increment change; its content increased tenfold. This compound is responsible for an almond-like odor [28]. The aldehydes can be produced through the degradation and oxidation of amino acids and reducing sugar during drying [29,30]. The high temperature in the drying is also conducive to the increase in the content of linalool, linalool oxides, and sulcatone compounds.

Linalool content increased during green tea production. It is the major volatile compound in the Chinese high-grade spring tea, Longjing tea [8], constituting 2.06%. In GT, linalool increased from 0.69% to 0.92%. This level was lower than that in spring tea. Large amounts of linalool are produced from terpenes during drying, and its content can be increased by increasing the drying time or drying temperature [19]. 

The content of ethyl acetate, butyl propionate, propyl butanoate, butyl acetate, methyl salicylate, and other ester compounds increased, which was related to the enhancement of esterification reaction in the process of degreening and frying. Under the action of heat, the original esters of FTL increased continuously, and new esters were produced. Among them, ethyl acetate and propyl butanoate, which have a soft fruit odor, and methyl salicylate, which has minty, wintergreen-like odors. These compounds are all related to the fruity and floral aroma of GT. 

### 3.2. Volatile Compounds Identified by GC-MS 

As shown in Table 2, a total of 104 volatile compounds were identified by GC-MS in the three samples, and 59, 59, and 57 volatiles were identified in FTL, RT, and GT, respectively. All these identified volatiles belong to nine categories; aldehydes, alcohols, ketones, esters, and alkenes were the major volatiles. Their proportions were 15.82%, 11.86%, 8.90%, 7.58%, and 14.89%, respectively, in FTL; 6.69%, 24.94%, 8.12%, 14.27%, and 8.42%, respectively, in BT; and 16.46%, 11.58%, 19.26%, 8.20%, and 3.56%, respectively, in GT. The number and proportions of the volatile compounds are shown in Figure 4. 

Aldehydes, alcohols, and alkenes were the major volatiles in FTLs. The most abundant compounds were d-limonene, hexanal, linalool, and phenylethyl alcohol. D-limonene was responsible for the fruit odor, and the other three were the vital aroma compounds imparting floral odor in tea [1]. Heptanal has a grassy odor, and methyl salicylate has a minty odor. They are the major contributors of green odor [31]. In addition, benzeneacetaldehyde, homocamphenilol, and β-ionone account for floral odors; (E,E)-2,4-hexadienal, benzeneacetaldehyde, and 2-pentyl-furan contribute to green odors; and some other ester compounds impart fruity odors [32] in FTL.

Alcohols and esters were the major categories in the BT sample, and 16 alcohol and 11 ester volatiles were identified. The proportions of alcohols and esters from FTL to BT both increased twofold. However, the proportion of aldehydes in BT decreased to 6.69%, less than half of that in FTL. The GT sample contained aldehydes, alcohols, and ketones as the major category. Compared with FTLs, GT showed higher amounts of ketones and esters, including 11 and 10 volatiles, respectively. The proportion of ketones from FTLs and GT increased twofold, but the proportion of alkenes in GT decreased to 3.56%, about a quarter of that in FTLs.

The content of aldehydes increased in GT, and decreased significantly in BT, because it was mainly produced by the degradation and oxidation of amino acids and reducing sugars [30]. The (E,E)-2,4-heptadienal, with nutty and green odors, increased significantly in GT, from 1.52% to 13.46%, and increased considerably during drying [1]. This result was consistent with the GC-IMS result. However, hexanal decreased in BT and GT, inconsistent with the GC-IMS results. This finding shows some differences between the results of the two experimental methods. The high content of compounds was detected mainly by GC-MS [13], and the low content of compounds were detected mainly by GC-IMS [11]. (E,E)-2,4-Decadienal, with cucumber and melon odors, was not detected in FTL or BT, evidently it was accumulated during GT producing.

Among alcohols, linalool and geraniol were abundant in BT and GT, similar to famous Chinese spring teas (like Keemun black tea [26] and Xihu Longjing tea [33]). The linalool oxides (I, II, III) formed during BT and GT production; increasing more significantly in BT during fermentation. These compounds are important sources of sweet and floral fragrances in BT. The content of phenylethyl alcohol, which has floral and rose-like odors, increased slightly in BT but significantly decreased in GT. Benzyl alcohol, with sweet and floral odors, and α-terpineol, with a floral odor, were detected only during GT production [17].

Ketones are generally considered to present tallow and burnt aromas, and also present floral aromas that are enhanced as the carbon chain grows [29]. In BT and GT, the content of β-Ionone increased significantly, and even more in GT, mainly because of the enzymatic oxidation of β-carotene during fermentation in BT processing or thermal degradation during GT processing [31]. The content of geranyl acetone increased in GT and decreased in BT. Sulcatone, with green and fruity odors, was detected in BT and GT, and 3, 5-octadien-2-one, with a rose-like odor, was only detected in GT. These ketones were probably produced by the oxidation and condensation of carotenoids, such as phytoalkenes and phytofluoroalkenes [29].

### 3.3. Key-Aroma Analysis by ROAV

Thousands of volatile compounds have been identified in food, but few have significant contributions to food flavor. These significant compounds are known as key aroma compounds [28]. ROAV can be calculated according to compound relative content and threshold value and used to evaluate the contributions of compounds to food flavor [21]. In previous studies [34], we found that black and green tea products made from Fuliang summer tea have high contents of flavor compounds (polyphenols, polysaccharides, and amino acids).

The ROAV values of 28 volatile compounds exhibited ROAVs (Table 3). Six compounds showed significant contribution (ROAV ≥ 1) in BT: hexanal (1.33), nonanal (1.79), decanal (15.63), linalool (100), geraniol (2.61), and β-ionone (1.65). This result indicated that the BT made from summer tea has the same key flavor compounds as Keemun black tea [25]. Linalool had the highest ROAV value. It has floral, fruit, green, and orange-like odors, and contributes to the flower and fruit aromas of BT. The second-highest was decanal, which has orange and sweet odors, and contributes to the aroma quality of BT due to its low threshold (0.1 μg/kg). Geraniol with rose-like, sweet, and honey-like odors, and β-ionone with floral, sweet, fruity, and berry-like odors, mainly contributed to the floral aromas in BT. Hexanal, with grassy, green, and fresh odors, and nonanal, with floral, green, and lemon-like odors, are fragrance aldehyde compounds and may have played a major role in the fragrance of BT. In addition, seven compounds were found to have important effects on the aroma quality of BT (0.1 ≤ ROAV < 1): dihydroactinidiolide (0.77), methyl salicylate (0.76), benzeneacetaldehyde (0.65), β-homocyclocitral (0.47), safranal (0.40), 2-pentyl-furan (0.21), and d-limonene (0.13). Benzeneacetaldehyde, with floral and green odors, showed the highest aroma intensity in Assam black tea, whereas methyl salicylate had the highest concentration in Ceylon black tea. Methyl salicylate is the typical aroma characteristic of Ceylon black tea [26].

Five compounds contributed significantly to the aroma of GT (ROAV ≥ 1): nonanal (1.98), decanal (14.09), (E,E)-2,4-decdienal (14.29), linalool (100), and β-ionone (3.12). Linalool had the highest ROAV value, which is responsible for the flower and fruit aromas, and is high in many kinds of high-quality spring green tea products, such as Jingshan tea [35], Japanese green tea [36], and Xihu Longjing tea [33]. The second was decanal, which had orange and sweet odors, and contributed significantly to the aroma quality of GT due to its low threshold. In addition, (E,E)-2,4-decadienal, with cucumber and melon odors, and nonanal, with floral, green, and lemon-like odors, may have contributed to the fragrance of GT. β-Ionone, which has floral, sweet, and fruity odors, can enhance the sweet characteristic of Japanese spring green tea [36]. In addition, eight other compounds were found to have important effects on the aroma quality of GT (0.1 ≤ ROAV < 1), including hexanal (0.98), heptanal (0.94), methyl salicylate (0.73), dihydroactinidiolide (0.68), geraniol (0.60), 2-pentyl-furan (0.26), sulcatone (0.16), and geranyl acetone (0.13). Geranyl acetone, which has a floral odor, is the odor-active compound in Japanese green tea and can be one of the markers for the overall quality evaluation of green tea [36]. Geraniol, which had rose-like and sweet odors, showed a high flavor dilution factor. It is the main aroma compound of Chinese green tea [5].

### 3.4. Combination of GC-IMS and HS-SPME-GC-MS

The Venn diagram (Figure 5) showed that 10–11 common compounds were detected in each sample through GC-IMS and GC-MS. And much more compounds were detected only in GC-IMS or GC-MS. And the flavor compounds detected by GC-IMS had 65 varieties in FTL, 64 varieties in BT, and 65 varieties in GT. The flavor compounds detected by GC-MS had 49 varieties in FTL, 48 varieties in BT, and 47 varieties in GT. Specifically, heptanal, hexanal, (E,E)-2,4-heptadienal, nonanal, decanal, linalool, and 2-pentyl-furan were all the flavor compounds detected in the three samples by both GC-IMS and GC-MS, which were speculated to be the key flavor compounds of teas due to the high ROAV value. Other key compounds in tea, including linalool oxide I, sulcatone and methyl salicylate, were detected in BT and GT, but not in FTL. This result indicated that the compounds accumulated during processing. 

The two methods have different detection principles. IMS technology separates ions according to the different migration rates when they pass through the gas in the electric field at ambient pressure [11], and MS technology separates ions according to the mass-to-charge ratio of the ions in the samples [13]. Most volatile compounds detected by GC-IMS are small molecules with low content, and most volatile compounds detected by GC-MS are large molecules with high content; therefore, GC-MS has low sensitivity to low concentrations. Consequently, the content value of compounds affects the results of the two detection methods, especially compounds with low content. Some key compounds with high content had the same change trends in tea aroma in the GC-MS and GC-IMS results. However, the results of the two methods are inconsistent on some compounds with low content, such as hexanal, nonanal, and decanal. Sampled and enriched by HS-SPME, volatile compounds detected by GC-MS present the same odors we can smell in practice. Therefore, we used GC-MS coupled with ROAV to analyze key aroma compounds in tea.

The varieties of flavor compounds detected in tea by only one method alone ranged from 47 to 65 kinds, which were far lower than the 122–124 kinds detected by the combination of the two methods. Therefore, the combination of the two methods can detect more material changes. The main flavor compounds detected only in GC-IMS were benzaldehyde, furfural, (E)-2-hexenol, 1-octen-3-one, ethyl acetate, and 2-ethyl pyrazine. The main flavor compounds detected only in GC-MS were safranal, β-homocyclocitral, benzylalcohol, phenylethyl alcohol, geraniol, geranyl acetone, β-ionone, and methyl salicylate. All these major flavor compounds can be detected simultaneously by combining the two methods.

The results of the methods were complementary with little difference. The compounds with high content were detected mainly by GC-MS, and the compounds with low content were detected mainly by GC-IMS. Therefore, the combination of the two technologies expanded the scope of volatile compounds detected in the samples in terms of content. The research reflected chemical changes in volatile compounds associated with the different tea processing methods comprehensively. By using a combination of the two methods, we can detect more compounds, in order to promote the quality of summer tea toward that of spring tea.

## 4. Conclusions

The aroma profiles of FTLs plucked in summer, and BT and GT made from them, were comprehensively analyzed by GC-MS, GC-IMS, and analyzed using ROAV. The study focused on changes of volatiles and their odor properties, which affect the flavor of teas. The results indicated that BT and GT made from fresh summer leaves have the same characteristic compounds as spring products, particularly linalool, geraniol, hexanal, and β-ionone. All these compounds have high content and ROAV value, which give the same flavor to summer teas as that of spring teas.

GC-IMS coupled with HS-SPME-GC-MS can monitor chemical changes in volatile compounds comprehensively and accurately. This provides a basis for adjusting processing parameters for improving summer tea quality and can thus promote the use of summer leaves in the production of valuable tea. Additionally, this method could be applied on spring tea and other tea products, monitoring the aroma compound changes during processing, and upgrading their quality.

## Figures and Tables

**Figure 1 foods-12-00146-f001:**
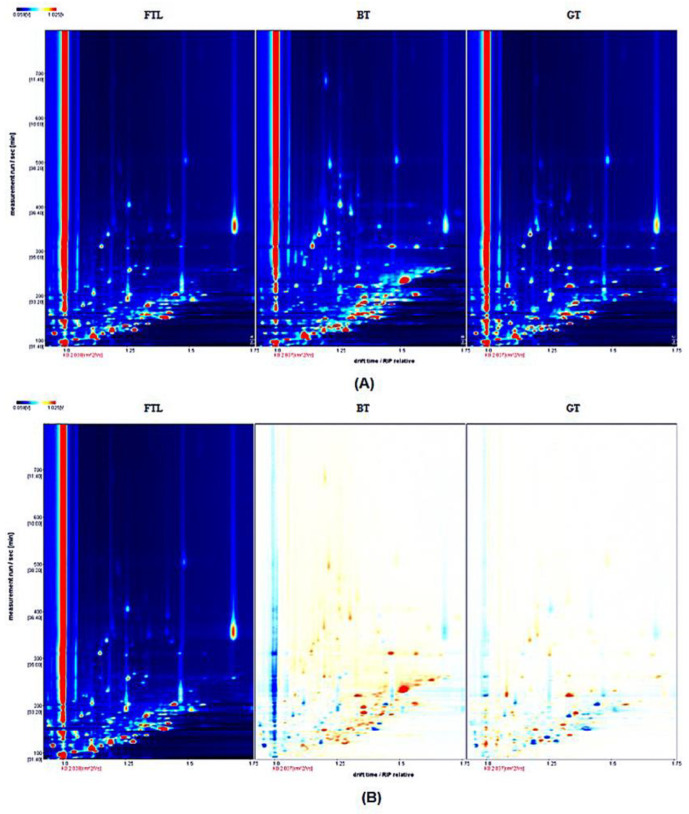
Topographic plots of three tea samples. (**A**) 2D-topographic plot and (**B**) differentiation plot of volatile compounds. In BT and GT, red and blue dots indicated that the concentration of the compounds are higher and lower than the reference (FTL), respectively.

**Figure 2 foods-12-00146-f002:**
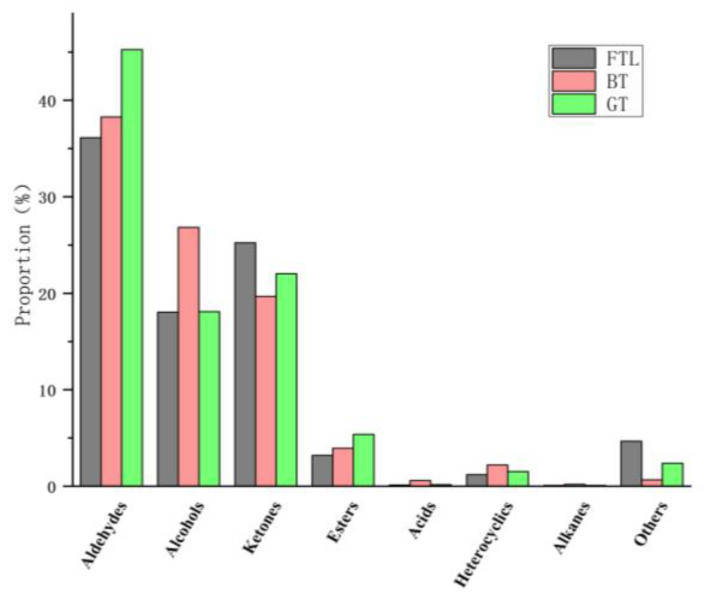
The proportion of volatiles of different categories identified by GC–IMS in three tea samples.

**Figure 3 foods-12-00146-f003:**
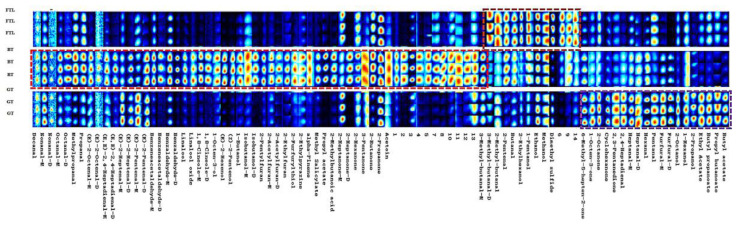
Gallery plot fingerprint of different teas by GC-IMS. The compounds with higher content are framed in the colored boxes: brown for FTL, red for BT, while/purple for GT.

**Figure 4 foods-12-00146-f004:**
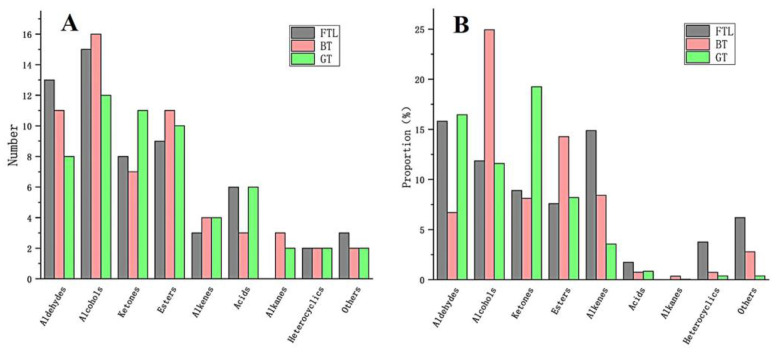
Volatile profiles of three tea samples. (**A**) The number of volatile compounds of different categories identified by GC-MS in three tea samples. (**B**) The proportion of volatiles of different categories identified by GC-MS of three tea samples.

**Figure 5 foods-12-00146-f005:**
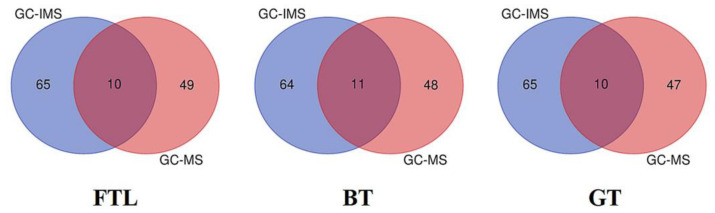
Venn diagram of the flavor of GC-MS and GC-IMS in different teas.

**Table 1 foods-12-00146-t001:** Volatile compounds identified in different teas by GC-IMS.

No.	Volatile Compounds	RI	Relative Content (%)			Odor Description ^#^
			FTL	BT	GT	
**Aldehydes**						
1	Decanal	1270.5	0.99	0.75	1.19	Orange, sweet
2	Nonanal-M	1112.1	1.85	1.61	2.70	Floral, green, lemon-like
3	Nonanal-D	1109.4	0.55	0.48	0.75	Floral, green, lemon-like
4	(E)-2-Octenal-M	1057.3	0.30	0.76	0.81	Green, nutty
5	(E)-2-Octenal-D	1055.1	0.18	0.19	0.21	Green, nutty
6	Benzeneacetaldehyde-M	1041.2	1.86	2.07	1.01	Floral, green, sweet
7	Benzeneacetaldehyde-D	1041.2	0.16	0.32	0.14	Floral, green, sweet
8	(E,E)-2,4-Heptadienal-M	1013.6	0.89	1.51	1.77	Nutty, green
9	(E,E)-2,4-Heptadienal-D	1013.6	0.19	0.40	0.29	Nutty, green
10	Octanal-M	1007.7	0.50	0.77	0.85	Fruity
11	Octanal-D	1007.7	0.14	0.25	0.23	Fruity
12	2,4-Heptadienal	1002.7	0.19	0.26	0.90	Fatty, nutty, hay, green, oily
13	Benzaldehyde-M	963.3	1.75	1.97	1.42	Almond, burnt sugar
14	Benzaldehyde-D	961.6	1.06	2.72	0.95	Almond, burnt sugar
15	(E)-2-Heptenal-M	955.3	0.58	0.45	1.28	Grass, cream
16	(E)-2-Heptenal-D	957	0.22	0.47	0.61	Grass, cream
17	Heptanal-M	902.9	0.74	0.73	1.32	Green, oily, grassy
18	Heptanal-D	903.6	0.17	0.29	0.63	Green, oily, grassy
19	Furfural-M	827.5	0.53	0.80	2.65	Sweet, bready, caramel
20	Furfural-D	826.8	0.30	2.02	5.18	Sweet, bready, caramel
21	Hexanal	792.5	1.10	2.55	2.58	Green, fresh, fatty
22	(E)-2-Pentenal-M	745.2	0.69	0.35	1.31	—
23	(E)-2-Pentenal-D	745.2	1.22	3.51	3.72	—
24	2-Pentenal	729.5	3.15	1.83	0.85	—
25	Pentanal	691.4	2.11	0.74	2.57	—
26	2-Methyl-butanal	647.7	3.56	1.18	2.26	Bitter apricot kernels
27	3-Methyl-butanal-D	627.4	3.50	0.90	1.22	Malty, alcohol
28	3-Methyl-butanal-M	626.8	0.48	1.72	2.17	Malty, alcohol
29	Butanal	572	1.18	0.67	0.56	—
30	Methyl propanal	514.5	4.23	4.66	1.86	—
31	Propanal	482.5	1.75	1.34	1.27	—
**sum**			**36.12**	**38.27**	**45.26**	
**Alcohols**						
1	Linalool	1105.1	0.69	1.67	0.92	floral, fruit, green, orange-like
2	Linalool oxide	1085.9	0.26	0.75	0.30	Sweet, floral, creamy
3	1,8-Cineole-M	1027.8	0.25	1.22	0.39	—
4	1,8-Cineole-D	1028.8	0.10	0.34	0.12	—
5	1-Octen-3-ol	985.6	0.29	0.52	0.38	Earthy, green, vegetative-like, fungal
6	2-Furfurylthiol	942	0.10	0.31	0.19	—
7	(E)-2-Hexenol	848.9	0.73	10.83	1.54	Green, fruit, vegetable
8	1-Pentanol	760.7	2.40	1.52	1.09	—
9	(Z)-2-Pentenol	777.5	0.21	1.83	0.51	—
10	1-Butanol	694.1	0.39	0.78	0.36	—
11	Isobutanol-M	600	0.28	0.25	0.17	—
12	Isobutanol-D	600.7	0.04	0.24	0.03	—
13	Ethanol	425.7	4.87	2.53	4.26	—
14	2-Ethylhexanol	1031.4	0.66	0.35	0.43	—
15	2-Octanol	989.4	0.50	0.17	0.79	—
16	1-Hexanol	881.7	0.51	0.38	0.63	—
17	Methanol	399.1	5.06	2.62	3.73	—
18	2-Propanol	523.7	0.70	0.52	2.26	—
**sum**			**18.04**	**26.83**	**18.10**	
**Ketones**						
1	Sulcatone	994.7	1.14	1.26	2.49	Green, fruity
2	2-Heptanone-M	895.3	1.59	0.66	1.29	Banana, cheese, fruit, medicinal
3	2-Heptanone-D	893.8	0.12	1.22	0.09	Banana, cheese, fruit, medicinal
4	2,3-Pentanedione	671.8	0.70	0.45	1.61	—
5	2-Butanone	560.2	4.96	4.24	1.44	—
6	2-Propanone	480.6	14.83	10.52	12.21	—
7	2-Octanone	1002.2	0.61	0.38	0.72	—
8	1-Octen-3-one	981.7	0.23	0.15	0.44	Earth, mushroom
9	Cyclohexanone	901.5	0.58	0.31	0.98	—
10	2-Hexanone	781.5	0.15	0.08	0.13	—
11	Acetoin	702.6	0.18	0.19	0.32	—
12	3-Pentanone	683.9	0.14	0.22	0.31	—
**sum**			**25.23**	**19.68**	**22.03**	
**Esters**						
1	Methyl Salicylate	1234.1	0.54	1.27	0.79	Minty, winter green-like
2	Propyl acetate	736.5	0.36	0.83	0.09	—
3	Ethyl Acetate	579.1	1.04	0.36	2.46	Fruit
4	Butyl propanoate	909.4	0.44	0.08	0.57	—
5	Propyl butanoate	898.1	0.39	0.14	0.73	Fruit
6	Butyl acetate	802.7	0.17	0.07	0.39	—
**sum**			**3.20**	**3.93**	**5.37**	
**Acids**						
1	2-Methylbutanoic acid	902.1	0.13	0.59	0.17	—
**sum**			**0.13**	**0.59**	**0.17**	
**Heterocyclics**						
1	2-Pentylfuran	997.5	0.31	0.61	0.35	Caramel, cooked, fruity, green
2	2-Ethylpyrazine	932.9	0.20	0.31	0.27	Nutty, potato, toasted, cocoa
3	2-Acetylfuran-M	912.8	0.34	0.62	0.54	Nutty, sweet, roasted, baked
4	2-Acetylfuran-D	912.4	0.15	0.36	0.17	Nutty, sweet, roasted, baked
5	2-Ethylfuran	693.1	0.21	0.30	0.19	Bean, fruity, earthy, green, vegetable
**sum**			**1.21**	**2.20**	**1.52**	
**Alkenes**						
1	*β*-Pinene	928.7	0.08	0.20	0.09	Turpentine, resin
**sum**			**0.08**	**0.20**	**0.09**	
**Others**						
1	Dimethyl sulfide	488	4.67	0.67	2.38	Delicate at low concentrations.
**sum**			**4.67**	**0.67**	**2.38**	

M, monomers; D, dimers. RI, retention index, which was calculated with reference to the retention time of C4–C9 n-ketones under the same conditions. # Odor description found in the literature (Flavornet; The LRI and Odour Database); —, no odor description information was found in the literature. The bold part represents the kind of the compound and the total relative content of each class.

**Table 2 foods-12-00146-t002:** Volatile compounds identified in different teas by GC-MS.

No.	Volatile Compounds	Relative Content (%)			Odor Description ^#^
		FTL	BT	GT	
**Aldehydes**					
1	Hexanal	4.87	1.91	0.97	Grassy, green, fresh, fatty
2	Heptanal	0.34	0.07	0.58	Green, oily, grassy
3	Octanal	0.68	ND	ND	Fruity
4	(E,E)-2,4-Heptadienal	1.52	1.65	13.46	nutty, green
5	(E)-2-Octenal	1.23	ND	ND	—
6	Benzeneacetaldehyde	2.28	0.83	ND	Floral, green, sweet
7	Nonanal	2	0.63	0.48	Floral, green, lemon-like
8	13-Tetradecenal	0.76	0.38	0.41	—
9	2-ethyl-Benzaldehyde	ND	ND	0.03	—
10	Decanal	0.8	0.5	0.31	Orange, sweet
11	safranal	0.07	0.38	ND	Herbal, fruit
12	β-Cyclocitral	0.96	ND	ND	Herbal, clean, rose-like, fruity
13	β-Homocyclocitral	0.3	0.03	ND	Camphor, cool wood
14	Citral	ND	0.13	ND	
15	(E,E)-2,4-Decadienal	ND	ND	0.22	Cucumber, melon
16	2-[4-methyl-6-(2,6,6-trimethylcyclohex-1-enyl)hexa-1,3,5-trienyl]cyclohex-1-en-1-carboxaldehyde	0.01	0.18	ND	—
**sum**		**15.82**	**6.69**	**16.46**	
**Alcohols**					
1	6-Methyl- bicyclo [4.2.0]octan-7-ol	0.28	0.15	ND	—
2	Linalool oxide I	ND	1.11	1.48	Sweet, floral, creamy
3	Benzyl alcohol	ND	ND	0.55	Sweet, floral, rose-like, caramel
4	1-[(1-Ethynylcyclohexyl)oxy]-2-propanol	0.03	ND	ND	—
5	Linalool oxide II	ND	1.99	ND	Sweet, floral, creamy
6	Linalool	2.2	7.07	4.86	Floral, fruit, green, orange-like
7	8-hydroxylinalool	ND	0.04	ND	—
8	trans-Carveol	ND	0.11	ND	—
9	Phenylethyl Alcohol	2.02	2.14	0.18	Floral, rose-like
10	α-Methyl-α-[4-methyl-3-pentenyl] oxiranemethanol	0.14	ND	ND	—
11	linalool oxideIII	ND	0.25	ND	Floral, honey-like
12	homocamphenilol	0.06	ND	ND	—
13	Homomyrtenol	ND	0.41	ND	—
14	α-terpineol	ND	ND	0.12	Pleasant, floral
15	(Z)- 3,7- dimethyl- isobutyrate -2-Octen-1-ol	ND	ND	0.26	—
16	13-Tetradecenal	2.14	ND	ND	—
17	Geraniol	ND	6.27	0.99	Rose-like, sweet, honey-like
18	2-methyl-4-(1,3,3- trimeth yl-7- oxa bicyclo [4.1.0] hept-2-yl)-3-Buten-2-ol	0.51	ND	ND	—
19	(6-Hydroxymethyl-2,3- dimethylphenyl)methanol	ND	0.88	1.12	—
20	humulol	ND	0.5	ND	—
21	1-Heptatriacotanol	0.62	0.62	0.61	—
22	2,2,6-Trimethyl-1-(3-methylbuta-1,3-dienyl)-7-oxabicyclo [4.1.0]heptan-3-ol	0.07	ND	ND	—
23	[1S-(1.alpha.,3.beta.,3a.beta.,4.alpha.,8a.beta.)]-1,4-decahydro-1,5,5,8atetramethyl -Methanoazulen-3-ol	0.03	ND	ND	—
24	2,2-Dimethyl-6-methylene-1-[3,5-dihydroxy-1-pentenyl]cyclohexan-1-perhydrol	0.08	0.17	0.05	—
25	tert-Hexadecanethiol	0.04	0.4	0.14	—
26	Isocalamenediol	0.44	ND	ND	—
27	shyobunol	ND	2.83	1.22	—
28	Olivetol	3.2	ND	ND	—
**sum**		**11.86**	**24.94**	**11.58**	
**Ketones**					
1	2-Nonadecanone	ND	ND	0.03	—
2	sulcatone	ND	0.81	1.75	Green, fruity
3	2,2,6-trimethyl-Cyclohexanone	0.11	ND	ND	—
4	Isophorone	0.1	ND	0.08	Cooling, woody, sweet, green, fruity
5	3,5-Octadien-2-one	ND	ND	2.91	Rose-like, lavender-like
6	3-Nonen-2-one	ND	0.04	ND	—
7	4-(2,2,6-Trimethylbicyclo [4.1.0]hept-1-yl)-2-butanon	0.12	ND	ND	—
8	5,9,9-trimethyl-Spiro [3.6] deca-5,7-dien-1-one	ND	ND	0.06	—
9	2-Hydroxycyclopentadecanone	ND	ND	0.06	—
10	4-(2,6,6-trimethyl-2-cyclohexen-1-yl)3-Buten-2-one	0.49	0.69	4.8	—
11	Geranyl acetone	1.44	1.2	1.68	Fresh, rose-like, floral, green, fruity
12	β-Ionone	2.79	4.44	5.76	Floral, woody, sweet, fruity, berry
13	4-(4-hydroxy-2,2,6-trimethyl-7-oxabicyclo [4.1.0]hept-1-yl)-3-Buten-2-one	2.45	0.9	1.24	—
14	1′-carboethoxy-1′-cyano-1,2-dihydro-3′H-Cycloprop(1,2)-5-cholest-1-en-3-one	1.4	0.04	0.89	—
**sum**		**8.9**	**8.12**	**19.26**	
**Esters**					
1	Linalyl acetate	ND	0.25	0.07	Fruity, floral
2	(Z)-verbenyl acetate	ND	0.33	ND	—
3	2-Methyl-3-methylene-1-cyclopentanecarboxylic acid methyl ester	0.66	ND	ND	—
4	E-2-Hexenyl benzoate	ND	ND	0.04	—
5	Acetic acid,3-(2,2- dimethyl-6-methylene- cyclohexylidene)-1-methyl-butyl ester	ND	ND	0.12	—
6	Terpinyl formate	0.27	ND	ND	—
7	Methyl salicylate	ND	9.75	6.45	Minty, wintergreen, grass odor
8	Formic acid, 3,7,11-trimethyl-1,6,10-dodecatrien-3-ylester	ND	0.27	ND	—
9	Terpinyl Acetate	ND	0.58	ND	—
10	Undec-10-ynoic acid, tridec-2-yn-1-ylester	0.26	ND	ND	—
11	2,5-Octadecadiynoicacid, methyl ester	ND	ND	0.21	—
12	Fumaric acid, 2-pentyl tridec-2-yn-1-yl ester	ND	0.09	ND	—
13	Undec-10-ynoic acid, dodecyl ester	ND	ND	0.02	—
14	Z-(13,14-Epoxy)tetradec-11-en-1-ol acetate	1.22	ND	ND	—
15	10-Methyl-8-tetradecen-1-olacetate	0.05	0.34	ND	—
16	Dasycarpidan-1-methanol, acetate	0.13	0.04	0.06	—
17	7-Methyl-Z-tetradecen-1-olacetate	0.38	0.58	0.28	—
18	Dihydroactinidiolide	2.46	0.94	0.57	Sweet, faint floral, herbal
19	Ethyl iso-allocholate	2.15	1.1	0.38	—
**sum**		**7.58**	**14.27**	**8.2**	
**Alkenes**					
1	Styrene	0.19	ND	0.45	Floral
2	3-Isopropyl-1-cyclohexene	ND	ND	0.21	—
3	D-Limonene	13.91	8.06	2.57	Citrus, lemon, orange-like, green
4	Carene	ND	0.04	ND	Woody
5	3-[(2-methylpropan-2-yl)oxy]bicyclo [3.2.1]oct-3-ene	ND	ND	0.33	—
6	Dipentene dioxide	ND	0.13	ND	—
7	Guaiene	0.79	0.19	ND	—
**sum**		**14.89**	**8.42**	**3.56**	
**Acids**					
1	Oleic Acid	0.92	0.58	0.31	—
2	3-Hydroxydodecanoic acid	0.02	ND	0.09	—
3	cis-7-Hexadecenoic acid	0.13	ND	ND	—
4	cis-4-(Hydroxymethyl) cyclohexanecarboxylic Acid	0.32	ND	ND	—
5	trans-13-Octadecenoicacid	0.22	ND	0.04	—
6	Ricinoleic acid	ND	0.06	ND	—
7	Linoleic acid	ND	ND	0.09	Oily
8	cis-5,8,11,14,17-Eicosapentaenoic acid	ND	ND	0.02	—
9	2-(3-acetoxy-4,4,14-trimethy landrost-8-en-17-yl)- Propanoic acid	0.13	0.11	0.29	—
**sum**		**1.74**	**0.75**	**0.84**	
**Alkanes**					
1	octyl-Oxirane	ND	0.24	ND	—
2	2-ethyl-1,1-dimethyl-Cyclopentane	ND	0.01	ND	—
3	1,2-15,16-Diepoxyhexadecane	ND	0.09	ND	—
4	4-(Hexadecyloxy)-2-pentadecyl-1,3-dioxane	ND	ND	0.01	—
5	2,6,10-trimethyl-Tetradecane	ND	ND	0.02	—
**sum**		**0**	**0.34**	**0.03**	
**Heterocyclics**					
1	2-amino-5-[(2-carboxy)vinyl]-Imidazole	2.89	0.33	0.04	—
2	2-pentyl-Furan	0.87	0.4	0.34	Bean, fruity, earthy, green, vegetable
**sum**		**3.76**	**0.73**	**0.38**	
**Others**					
1	*p*-Cymene	1.87	ND	ND	Aromatic
2	Geranyl vinyl ether	ND	0.6	ND	—
3	Caffeine	4.24	2.18	0.33	Caffeine
4	1,2-Dimyristoyl-sn-glycero-3-phosphocholine	0.09	ND	0.05	—
**sum**		**6.2**	**2.78**	**0.38**	

# Odor description found in the literature (Flavornet; The LRI and Odour Database); —, no odor description information was found in the literature; ND, not detectable. Others, the volatile compounds were not from aldehydes, alcohols, ketones, esters, acids, hydrocarbons, and alkenes compounds. The bold part represents the kind of the compound and the total relative content of each class.

**Table 3 foods-12-00146-t003:** The ROAV values of the main volatile compounds by GC-MS.

No.	Volatile Compounds	Odor Description ^#^	Thresold ^ψ^ (μg/L)	ROAV		
				FTL	BT	GT
1	Hexanal	Grassy, green, fresh, fatty	4.5	10.82	1.33	0.98
2	Heptanal	Green, oily, grassy	2.8	1.21	0.08	0.94
3	Octanal	Fruity	0.7	9.71	<0.01	<0.01
4	(E,E)-2,4-Heptadienal	Fatty, green	10,000	<0.01	<0.01	<0.01
5	Benzeneacetaldehyde	Floral, green, sweet	4	5.70	0.65	<0.01
6	Nonanal	Floral, green, lemon-like	1.1	18.18	1.79	1.98
7	Decanal	Orange, sweet	0.1	80.00	15.63	14.09
8	Safranal	Herbal, fruit	3	0.23	0.40	<0.01
9	β-Cyclocitral	Herbal, clean, rose-like, fruity	3	3.20	<0.01	<0.01
10	β-Homocyclocitral	Camphor, cool wood	0.2	15.00	0.47	<0.01
11	(E,E)-2,4-Decadienal	Cucumber, melon	0.07	<0.01	<0.01	14.29
12	Linalool oxide I	Sweet, floral, creamy	190	<0.01	0.02	0.04
13	Benzyl alcohol	Sweet, floral, rose-like, caramel	20,000	<0.01	<0.01	<0.01
14	Linalool oxide II	Sweet, floral, creamy	190	<0.01	0.03	<0.01
15	Linalool	floral, fruit, green, orange-like	0.22	100	100	100
16	Phenylethyl Alcohol	Floral, rose-like	390	0.05	0.02	<0.01
17	linalool oxide III	Floral, honey-like	3000	<0.01	<0.01	<0.01
18	α-terpineol	Pleasant, floral	330	<0.01	<0.01	<0.01
19	Geraniol	Rose-like, sweet, honey-like	7.5	<0.01	2.61	0.60
20	Sulcatone	Green, fruity	50	<0.01	0.05	0.16
21	Isophorone	Cooling, woody, sweet, green, fruity	11	0.09	<0.01	0.03
22	Geranyl acetone	Fresh, rose-like, floral, green, fruity	60	0.24	0.06	0.13
23	β-Ionone	Floral, woody, sweet, fruity, berry	8.4	3.32	1.65	3.12
24	Methyl salicylate	Minty, wintergreen-like, grass	40	<0.01	0.76	0.73
25	Dihydroactinidiolide	Sweet, faint floral, herbal	3.8	6.47	0.77	0.68
26	D-Limonene	Citrus, lemon, orange-like, green	200	0.70	0.13	0.06
27	2-pentyl-Furan	Bean, fruity, earthy, green, vegetable	6	1.45	0.21	0.26
28	p-Cymene	Aromatic	11.4	1.64	<0.01	<0.01

# Odor description found in the literature (Flavornet; The LRI and Odour Database). ψ All odor thresholds were obtained from: Odour & Flavour Detection Thresholds in Water (In Parts per Billion, μg/L).

## Data Availability

Data is contained within the article.

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
