# Peer review of "Monitoring Changes in the Volatile Compounds of Tea Made from Summer Tea Leaves by GC-IMS and HS-SPME-GC-MS"

_foods, 2022, doi:10.3390/foods12010146_

Round 1

Reviewer 1 Report

Line 393-394: What are the units for  '47-65' and '122-124' in the sentence as; The varieties of flavor compounds detected in tea by only one method alone ranged from 47 to 65, which were far lower than the 122-124 detected by the combination of the two methods. It should be explained.

Author Response

  • Thank you for this In first sentence, '47-65' and '122-124' refer to the number of volatile compounds detected in the tea sample. So, the units are kinds.
  • The second sentence refers to that when only GC-IMS or GC-MS was used,47 kinds of substances, or at the highest 65 kinds of substances could be detected. When the two methods were used together, as many as 122-124 kinds of substances could be detected.
  • In the revision, we added the unit‘kinds’after the number, seen in Line 432.

Reviewer 2 Report

Monitoring changes in the volatile compounds of tea made from summer tea leaves by GC-IMS and HS-SPME-GC-MS

In this study, the volatile compounds of black tea (BT) and green tea (GT) made from summer fresh tea leaves (FTLs) have been quantified using gas chromatography-ion mobility spectrometry (GC-IMS) and head space solid phase micro-extraction-gas chromatography-mass spectrometry (HS-SPME-GC-MS) and then analyzed on the basis of relative odor activity value (ROAV).

Comments:

The idea of the manuscript is quite interesting in food science and technology, the manuscript has been well written and the experimental and the aim of the manuscript is quite clear. However minor issues should be considered before publication:

-       The manuscript should be improved from the grammatical point of view.

-       Literature review about the application of GC-MS and GC-IMS for tea authentication should be improved through the Introduction.

-       For the next research activities of the authors I propose to use chemometrics methods which could reveal valuable information from too large data points obtained from the analytical methods such as GC-MS.

Considering the above issues, publication of the manuscript is recommended after minor revision.

Author Response

  1. The manuscript should be improved from the grammatical point of view.
  • Thank you for your reminding.The manuscript had been polished by professional language editor. And we also examined the language carefully, 
  • The certification of English editing was attached.
  1. Literature review about the application of GC-MS and GC-IMS for tea authentication should be improved through the Introduction.
  • Thank you for your reminding.In the revision, we added the sentence: In recent years, GC-MS and GC-IMS are often used in tea flavor analysis. Seen in Line 70-71.
  1. For the next research activities of the authors I propose to use chemometrics methods which could reveal valuable information from too large data points obtained from the analytical methods such as GC-MS.
  • Thank you for valuable suggestion. Inthe next research activities, we will consider to use chemometrics method.

Reviewer 3 Report

My comments are as follow:

·         In page 1 lines from 28 to 30. Since some words are repeated in the manuscript title and in the keyword section, I suggest to replace the words: “volatile compounds”; “summer tea” or “HS-SPME-GC-MS”; by “aroma compounds”; “Fuliang castanopsis” or other keywords, to have more visibility of your manuscript paper.

·         In the conclusion section, you stated that “This study provides a basis for adjusting processing parameters for improving summer tea quality and can thus promote the use of summer leaves for the production of valuable tea”. Nevertheless, what about the future work in your research? I suggest to state with more accurate your future research related with this study.

Author Response

  1. In page 1 lines from 28 to 30. Since some words are repeated in the manuscript title and in the keyword section, I suggest to replace the words: "volatile compounds"; "summer tea”or "HS-SPME-GC-MS"; by "aroma compounds"; "Fuliang castanopsis"or other keywords, to have more visibility of your manuscript paper.
  • Thanks for your valuable suggestion.In the revision, we added two key words "aroma compounds" and "Fuliang castanopsis" and deleted "volatile compounds", seen in Line 30.
  1. In theconclusion section, you stated that“This study provides a basis for adjusting processing parameters for improving summer tea quality and can thus promote the use of summer leaves for the production of valuable tea”. Nevertheless, what about the future work in your research? I suggest to state with more accurate your future research related with this study.
  • Thanks for reminding. In the revision, we added these contents: Additionally, this method could be applied on spring tea and other tea products, monitoring the aroma compounds change during processing and upgrading their quality.Seen in Line 459-461.

Reviewer 4 Report

This manuscript requires significant editing, as it is not written in sound English. The language mistakes are too many to mention. I strongly recommend to revise the language from a native speaker.

The novelty of the paper needs to be clearly stated. What are the contributions of this paper? Also need to discuss the difference between this work and previously conducted research in literature.

The title of the paper is not impressive. The title should be revised.

The abstract section needs to be rewritten and must include important findings.

Most figures are not clearly presented. The quality of the figures should be improved.

The conclusion section is weakly formulated. It should be improved with the inclusion of the most important research outcomes. So, suggest its needs to be revised and should give some future insight into it.

Author Response

  1. This manuscript requires significant editing, as it is not written in sound English. The language mistakes are too many to mention. I strongly recommend to revise the language from native speaker.
  • Thank you for your reminding.The manuscript had been polished by professional language editor. And we also examined the language carefully, 
  • The certification of English editing was attached.
  1. The novelty of the paper needs to be clearly stated. What are the contributions of this paper? Also need to discuss the difference between this work and previously conducted research in literature.
  • Thank you for your reminding. We have made suggested changes as below in the  
  • The novelty of the paper is explained in the introduction. At present, the research on tea aroma is mainly focused on spring tea, and relatively little research has been done on summer tea. In this paper, summer tea with large yield and serious waste was studied by GC-MS, GC-IMS, and analyzed using ROAV. Also, it clarified the difference between this work and previously conducted research. Seen in Lines 57-61, and 74-80, etc.
  • The contributions of this paper wereclarified in the conclusion. It confirmed that tea leaves plucked in summer was suitable to produce black and green tea with good flavor, which broadened the tea categories. GC-IMS coupled with HS-SPME-GC-MS can monitor compounds changes in volatile compounds comprehensively and accurately. This method could be applied on spring tea and other tea products, monitoring the aroma compounds change during processing and upgrading their quality. Seen in Line 456-461.
  • As seen in 1), in the novelty explain, we listed the differences.
  1. The title of the paper is not impressive. The title should be revised.
  • Thank you for your reminding. We think this title can cover the content of this literature.It is meaningful, although might be not impressive.
  1. Abstract section needs to be rewritten and must include important findings.
  • Thank you for your reminding.In the revision, the abstract had been modified. Added some important findings. Such as some important compounds content and the ROAV value are high, so that summer tea and spring tea have the same characterized  The Fuliang castanopsis summer leaves are suitable for making black and green tea with good flavor. Seen in Line 15-29.
  1. Most figures are not clearly presented. Quality of figures should be improved.
  • Thank you for your reminding.In the revision, we used all the original figures to ensure that they are clear enough.
  1. Conclusion section is weakly formulated. It should be improved with the inclusion of most important research outcomes. So, suggest its needs to be revised and should give some future insight in it.
  • Thank you for your reminding.In the revision, we added the sentence: Additionally, this method could be applied on spring tea and other tea products, monitoring the aroma compounds change during processing and upgrading their quality. Seen in Line 457-459.

Reviewer 5 Report

The manuscript "Monitoring changes in the volatile compounds of tea made from summer tea leaves by GC-IMS and HS-SPME-GC-MS" is an investigation into volatile profiles and odor characteristics of black tea and green tea prepared from Fuliang castanopsis summer fresh tea leaves. Through the use of GC-MS and GC-IMS, the authors reported that the two teas were composed of some volatile constituents similar to those of spring tea. In addition, it is concluded that the summer tea leaves were suitable for manufacturing green and black tea. Several points that the authors should need to make clarification of include:

- To avoid any confusion to readers, the authors should mention the scientific name of the summer fresh tea used in the study. Is it Camellia sinensis?

- In Section 2.2, the authors mentioned built-in NIST 2014 gas-phase retention index database was used. To clarify a bit more, were the retention indices of the reported compounds in the study calculated using an n-alkane series? If yes, please add this information in study (probably in Supplementary materials). Without this, the misidentification of volatile organic compounds can be inevitable.

Author Response

Response:

Reviewer 5

Comments and Suggestions for Authors

The manuscript "Monitoring changes in the volatile compounds of tea made from summer tea leaves by GC-IMS and HS-SPME-GC-MS" is an investigation into volatile profiles and odor characteristics of black tea and green tea prepared from Fuliang castanopsis summer fresh tea leaves. Through the use of GC-MS and GC-IMS, the authors reported that the two teas were composed of some volatile constituents similar to those of spring tea. In addition, it is concluded that the summer tea leaves were suitable for manufacturing green and black tea. Several points that the authors should need to make clarification of include:

  1. To avoid any confusion to readers, the authors should mention the scientific name of the summer fresh tea used in the study. Is it Camellia sinensis?
  • Thank you for this suggestion. Its scientific name is Camellia sinensis.
  • We use summer fresh tea leaves ( sinensis, cv. Fuliangzhong) to make this clearer. Seen in Line 78 and 84.
  1. In Section 2.2, the authors mentioned built-in NIST 2014 gas-phase retention index database was used. To clarify a bit more, were the retention indices of the reported compounds in the study calculated using an n-alkane series? If yes, please add this information in study (probably in Supplementary materials). Without this, the misidentification of volatile organic compounds can be inevitable.
  • Thank you for your reminding. We used the C4-C9 n-ketones to calculate retention indices of the reported compounds. In the revision, we added the sentence: C4-C9 n-ketones (Sinopharm Chemical Reagent Beijing Co., Ltd, China) were used as external references to allow the retention index (RI) of the detected volatiles to be calculated under the same chromatographic conditions. Seen in Line 113-116.
  • In table 2, we also added the retention indices of the reported compounds.

Round 2

Reviewer 4 Report

Accepted 

Author Response

Thank you very much

Reviewer 5 Report

Thank you for the authors' endeavor to clarify the questions that have been raised. 

Table 2 shows the retention indices that were calculated using n-alkane series and I assume these RI can be found in the built-in NIST 2014 gas-phase retention index database. The authors also previously mentioned the use of C4-C9 ketones to calculate RI. I am confused if these two are compatible, and if so they should be added in Table 2 too.

Author Response

Table 2 shows the retention indices that were calculated using n-alkane series and I assume these RI can be found in the built-in NIST 2014 gas-phase retention index database. The authors also previously mentioned the use of C4-C9 ketones to calculate RI. I am confused if these two are compatible, and if so they should be added in Table 2 too.

  • Thank you for reminding. In Line 202, We miswrote C4-C9 ketonesas C5-C28 alkane. In the revision, We have rewritten them as C4-C9 ketones.
  • The RI calculated using C4-C9 ketones is also applicable for this build-in NIST 2014 gas-phaseretention index database.